# Survival status and predictors of neonatal mortality among neonates admitted to Neonatal Intensive care Unit (NICU) of Wollega University referral hospital (WURH) and Nekemte Specialized hospital, Western Ethiopia: A prospective cohort study

Tadesse Tolossa[1]*, Bizuneh Wakuma[2], Belayneh Mengist[3], Getahun Fetensa[2], Diriba Mulisa[2], Diriba Ayala[4], Ilili Feyisa[1], Ginenus Fekadu[5,6], Dube Jara[3], Haile Bikila[1], Ayantu Getahun[1]

1 Department of Public Health, Institute of Health Science, Wollega University, Nekemte, Ethiopia,
2 Department of Nursing, Institute of Health Science, Wollega University, Nekemte, Ethiopia, 3 Department of Public Health, College of Health Science, Debre Markos University, Debre Markos, Ethiopia, 4 Department of Midwifery, Institute of Health Science, Wollega University, Nekemte, Ethiopia, 5 Department of Pharmacy, Institute of Health Science, Wollega University, Nekemte, Ethiopia, 6 School of Pharmacy, Faculty of Medicine, The Chinese University of Hong Kong, Shatin, New Territories, Hong Kong

* yadanotolasa@gmail.com

## Abstract

### Background

The neonatal period is the most vulnerable time for survival in which children face the highest risk of dying in their lives. Neonatal mortality (NM) remains a global public concern, especially in sub-Saharan African (SSA) countries. Although, better progress has been made in reducing NM before 2016, Ethiopia is currently one of the top ten countries affected by NM. Studies are limited to secondary data extraction in Ethiopia which focus only on survival status during admission, and no study has been conducted in the study area in particular.

### Objective

To assess the survival status and predictors of neonatal mortality among neonates admitted to the NICU of WURH and Nekemte Specialized Hospital, Western Ethiopia.

### Methods

An institution-based prospective cohort study was conducted among a cohort of 412 neonates admitted to the NICU of WURH and Nekemte Specialized Hospital from September 1, 2020 to December 30, 2020. All neonates consecutively admitted to the NICU of the two hospitals during the study period were included in the study. Data entry was performed using Epidata version 3.0 and the analysis was performed using STATA version 14. A Kaplan Meier survival curve was constructed to estimate the cumulative survival probability.

**Data Availability Statement:** All relevant data are within the paper and its Supporting information files.

**Funding:** The author(s) received no specific funding for this work.

**Competing interests:** The authors have declared that no competing interests exist.

**Abbreviations: AHR**, Adjusted Hazard Ratio; **ANC**, Antenatal Care; **CHR**, Crude Hazard Ratio; **EDHS**, Ethiopia Demographic and Health Survey; **NM**, Neonatal Mortality; **PDO**, Person Day Observation; **SSA**, Sub-Saharan Africa; **UNICEF**, United Nations Children's Fund; **WHO**, World Health Organization; **WURH**, Wollega University Referral Hospital.

A cox proportional hazards regression model was used to identify the predictors of NM. Hazard Ratios with 95% CI were computed and all the predictors associated with the outcome variable at p-value $\leq$ 0.05 in the multivariable cox proportional hazards analysis were declared as a significant predictor of NM.

## Results

A total of 412 neonates were followed for a median of 27 days with an IQR of 22–28 days. During the follow-up period, a total of 9249 person day observations (PDO) were detected. At the end of follow-up, 15.3% of neonates died with an overall incidence rate of death 6.81/ 1000 PDO. The median time to death was 10 days, and the highest incidence rate of death was observed during the first week of the neonatal period. The study found that rural residence (AHR = 2.04, 95%CI: 1.14, 3.66), lack of ANC visits (AHR = 7.77, 95%CI: 3.99, 15.11), neonatal hypothermia (AHR = 3.04, 95%CI: 1.36, 6.80), and delayed initiation of breastfeeding (AHR = 2.26, 95% CI: 1.12, 4.56) as independent predictors of NM. However, a decreased number of pregnancies decrease the risk of NM.

## Conclusions and recommendations

The incidence rate of neonatal death was high particularly in the first week of life in the study area. The study found that lack of ANC visit, neonatal hypothermia, increased number of pregnancies, rural residence, and delayed initiation of breastfeeding positively predicted NM. Therefore, there is a need to encourage programs that enhance ANC visits for pregnant mothers and community-based neonatal survival strategies, particularly for countryside mothers.

## Introduction

Neonatal mortality (NM) is defined as the death of a baby during the neonatal period (within the first 28 days of life) which is the most vulnerable time for survival in which children face the highest risk of dying in their life [1]. Although, a being newborn is not a disease, large numbers of children die soon after birth: many of them die in the first month and most of which occur in the first week of life. Newborns die after birth because they encounter difficulties in adapting to extra uterine life [2].

Globally, an average rate of 17 deaths per 1,000 live births was reported and 2.4 million children died in the first month of life in 2019 [3]. According to the World Health Organization (WHO) 2018 report, 40 neonatal deaths per 1000 live births were recorded [4]. Most neonatal deaths occur during the early neonatal period. Nearly 6,700 neonates die every day, of which approximately one third and three-quarters of all neonatal deaths occur within the first day after birth, and within the first week of life respectively [3]. The main causes of newborn death in the first month of life are premature birth, complications during labor and delivery, and infections acquired during the delivery process [5].

A marked disparity in NM rate is observed across regions and countries. Neonatal mortality was highest in SSA and South Asia. An estimated 27 deaths per 1,000 live births were reported in SSA, and a child born in SSA was 10 times more likely to die in the first month of life than a child born in a high-income country [3].

According to the 2019 Mini Demographic and Health Survey (MEDHS) report, 29 neonatal deaths per 1000 live births were reported in Ethiopia which is among the highest in sub-Saharan African countries [6]. Several factors contribute to neonatal deaths including poor maternal health, inadequate obstetric care; inappropriate management of complications during pregnancy, delivery, and post natal period, and lack of proper newborn care [2, 7]. In addition, women's status in society, nutritional status, non-spaced pregnancies, and harmful traditional practices such as inadequate cord care, discarding colostrum and feeding on other foods contribute to neonatal mortality [2]. Prematurity, low birth weight and infection are the main causes of neonatal mortality particularly in low and middle-income countries such as Ethiopia [8–10].

Significant progress has been made in reducing neonatal mortality since 1990 towards the achievement of Millennium Development Goals (MDGs) despite the progress remains stagnant particularly in SSA. The global neonatal mortality rate declined by 40 percent from 33 deaths per 1,000 live births in 1990 to 20 in 2013 [11].

In Ethiopia neonatal mortality decreased from 39 in 2005 to 29 in 2016 although an insufficient reduction was reported after 2016 [6]. Despite this progress, NM remains a global public concern, especially in SSA countries. The Ambitious Sustainable Development Goals (SDGs) targeted to end preventable deaths of newborns and children under 5 years of age with the aim of reducing neonatal mortality to at least 12 per 1000 live births by the end of 2030.

The majority of the causes of NM are preventable and the variation in neonatal mortality rate is due to the variations in the quality of care provided at neonatal intensive care units. Limited studies have been conducted on survival status and predictors of neonatal mortality in Ethiopia [1, 7], but these studies were conducted retrospectively, and did not include potential predictors of NM. In addition previous studies were only conducted when the neonates were in the admission unit by overlooking the survival status after the neonates discharged from hospital. A study using a strong design with long follow up for 28 days is required to produce sound evidence to reduce NM in resource limited settings. Therefore this study aimed to assess survival status and predictors of neonatal mortality among neonates admitted to the NICU WURH and Nekemte Specialized Hospital, Western Ethiopia.

## Methods

### Study area and period

This study was conducted in Hospitals found in Nekemte town (Nekemte specialized hospital and WURH), Oromia regional state, Western Ethiopia. These two hospitals provide a delivery service and also have NICU wards. There are different private and public health institutions found in the town. Generally, there are two health centers, one specialized hospital (Nekemte specialized hospital), and one referral hospital (WURH) in Nekemte town. The study period was from September 1, 2020 to December 30, 2020 among a cohort of consecutive neonates admitted to NICU at the selected hospitals with a maximum follow-up period of 28 days.

### Study design, population, and eligibility criteria

A four-month institution-based prospective cohort study was employed. All neonates who were admitted to the NICU in WURH and Nekemte specialized hospital were the source population and the study population included all neonates who were consecutively admitted to the NICU at selected Hospitals from September 1, 2020 to December 30, 2020. Neonates who were admitted to the NICU on the first day of delivery in the selected hospitals during the study period and their respective mothers or caregivers available during data collection were included in the study. Mothers who were not able to speak, or have psychiatric illnesses were

excluded from the study. Moreover, neonates who were delivered outside the selected hospitals including home delivery and admitted to the NICU after one day of delivery were excluded from the study because it was difficult to retrieve important baseline data immediately after delivery.

## Sample size and sampling techniques

To determine the representativeness of the population, the sample size for the cohort study was estimated using the double population proportion formula in EPI INFO version 7 by considering different significant variables from the previous study [12], and by considering the following assumptions; the level of significance was 5% and the power 80% P1 = Proportion of neonatal mortality among exposed and P2 = Proportion outcome among the non-exposed group. The ratio of the population exposed to non-exposed was 1:1. Variables such as lack of neonatal complications, sex of the baby, birth weight, postnatal care, and initiation of breastfeeding were used to calculate the sample size. Finally, the variable initiation of breastfeeding was selected for final sample size estimation which gave a sample size of 362 (P1 = proportion of neonatal mortality among neonates initiated breastfeeding after 1 hour was 87.0% and P2 = proportion neonatal mortality among neonates who initiated breastfeeding with in 1 hour was 96.0%). Then, 10% non-response was added to the estimated sample size, yielding a final sample size of 398.

## Variables and outcome measurements

The outcome variable of this study was neonatal mortality and which was the event of the study. Neonatal mortality is the death of neonates during the first 28 days after birth [13]. An outcome that did not develop an event [14] were recorded as censored. Censored includes neonates who survived the first 28 completed days after birth and alive at the end of the study, lost from follow up, and transferred to other health institutions. Survival time was defined as the time in days from the date of admission to the NICU to the occurrence of outcome (event/censored).

Several variables were considered in this study to investigate the major predictors of neonatal death as independent variables. The predictor variables considered for this study were; Socio-demographic factors such as residence (urban/rural), age of mother, educational status of mother (unable to read and write, primary, 9–12 grade, college and above), marital status (married, single, widowed, divorced), occupational status of mother (house wife/farmer, own business, government employee, student), ethnicity (Oromo, Amhara, Ghurage and others), religion (orthodox, protestant, and muslim), and distance from home to health institution.

Maternal related factors such as gestational age (preterm <259 days, term or 259–293 days, post term birth ≥294 days), body mass index (underweight or less than 18.5kg/m$^2$, normal 18.5–24.9kg/m$^2$, overweight/≥25 kg/m$^2$), number of pregnancy (no pregnancy, 1–2, 3–4, ≥5), ANC follow up (No ANC follow-up, 1–3, ≥4 visit), medical disease (Yes/No), place of delivery (home/ health institution), mode of delivery (spontaneous vaginal delivery, operative (instrumental) vaginal delivery and cesarean section), attendant of delivery (health professional, TBA), type of birth (multiple or single), preceding birth interval. Neonatal related factors such as birth weight (<2500gm, 2500–3900gm, and ≥ 4000gm), sex of neonates (male, female), and age at admission (≤ 1 day, >1 day) was also considered. Immediate (interventional) factors like temperature of neonates (<36.5, 36.5–37.5, >37.5), early initiation of breast feeding (≥ 1 hour, <1 hour), maternal complications (obstetric hemorrhage, puerperal sepsis, prolonged labor, eclampsia and preeclampsia, mal-presentation and mal-position, premature rupture of membrane (PROM), cord prolapse, obstructed labor), neonatal complication (asphyxia,

prematurity, infection, congenital anomaly, jaundice), pregnancy complications (vaginal bleeding, abdominal pain, persistence of back pain, blurry vision, no fetal movement and swelling of hands or face) was also included. Birth asphyxia is the failure to initiate and sustain breathing at birth and a newborn was considered to have birth asphyxia when its fifth minute APGAR score was <7 [15].

## Data collection procedure

Data were collected from mothers of the newborn and newborn using structured questionnaires, which was adopted from similar literature conducted in different settings [9, 12, 16], and Mosley and Chen conceptual framework previously used in similar studies [17, 18]. The questionnaires were consisting of socio-demographic data, maternal and neonatal-related factors, and interventional factors.

The questionnaire was initially developed in English and translated into the local language, Afaan Oromo, and back to English to ensure its consistency. The data collectors collect information by interviewing all mothers who delivered a live birth at the selected hospitals and their newborns were admitted to NICU. The baseline information of newborn was either extracted from the medical record which recorded by health workers in the hospital or collected from mother or newborn directly by data collectors. Data were collected by two midwives and two nurses working in the delivery room and NICU ward of selected hospital. Nurses and midwives collected baseline data, and data collectors (midwives and nurses) contact the mother and newborn every day to identify the outcome. The survival status of neonates after discharge was checked by making follow-up phone calls every day up to the end of the neonatal period, and followed by health extension workers every week. When an event or death happened, the date of death and important information were recorded on the checklist. Health extension workers visited those who were not be reached by phone every week to ascertain the status of the neonate or to know the outcome at the end of follow up. If an event or death happened at home, data related to the illness was recorded using the standard WHO verbal autopsy questionnaire [17]. Then it was supplemented with other variables from the hospital, which related to delivery and discharge information to provide additional information on the predictors of death. The principal investigator monitored the progress of data collection every day. Finally, if the mother's phone was not working or moves to another area from the original place of residence and was not met by health extension worker the outcome was recorded as LTFU.

To assure the quality of data, the questionnaire was pretested on the 5% of sample size at Gimbi general hospital, and the possible amendment was made. Four BSc Nurses and midwives, and two health extension worker was recruited for data collection and one BSc MPH supervised the overall data collection. The one-day training was given for data collectors and supervisors by principal investigator. During the data collection, the questionnaire was checked for the completeness and consistency by the principal investigator every day.

## Data management and analysis

Epidata version 3.2 was used for data entry, and then the data was exported to STATA version 14 for further analysis. Data were cleaned and edited by simple frequencies and cross-tabulation before analysis. Descriptive survival analysis such as Kaplan-Meier survival function estimation was used for the estimation of the distribution of survival time. Survival time was defined as the time in days from the date of admission to NICU to the occurrence of outcome (event or censored) within 28 days of live birth. Person-days of observation were calculated as; date of occurrence of an event or censored subtracted from the date of admission to NICU

ward to determine total days of follow up for all subjects under the study. Kaplan Meier survival curve together with log-rank test was fitted to test for the presence of difference in the occurrence of death among the covariates. The overall survival function and separate estimates for the stratum of covariates were considered as statistically significant at a p-value of 0.05 in the Log-rank test. Cox proportional hazards regression model was used to determine predictors of mortality by controlling confounding. Factors that were associated with outcome variables at 25% (p<0.25) significant level in the bivariable test were included in the final Multivariable analysis. Hazard Ratios (HR) with 95% confidence intervals were computed and statistical significance was declared when it was significant at the 5% level (p < 0.05). The model was fitted using backward selection among variables and a log likely hood ratio was used to select the best model. For categorical covariates, proportionality hazard assumption was tested graphically (log-log plot) and global goodness of fit test or Schoenfeld residuals were used to test proportionality hazard assumption for both continuous and categorical covariates. Cox–Snell residual plot was used to assess the overall goodness of fit of the proportional hazard model.

## Ethical considerations

Ethical clearance was obtained from the Review Ethics Committee of Wollega University, Institute of Health Sciences. A permission letter was written from the respective hospital administrative body to NICU wards. Oral informed consent was obtained from the mother after a clear explanation of the purpose of the study. Moreover, no personal identifiers were used on data collection questionnaires. The recorded data was not be accessed by a third person except the principal investigator and was kept secure.

## Results

### Maternal socio-demographic characteristics

A total of 412 mother-neonates cohorts participated in the study. One hundred fifty (36.4%) of mothers were found in age groups of 21–25 years old. More than half (60.4%) of mothers reside in urban settings, and more than 80% of the mothers were married. Eighty-three (20.1%) of the mothers are unable to read and write and more than 85% of the mothers were housewives (Table 1).

### Obstetric characteristics and maternal complications

Around one-fifth (23.5%) of the mothers gave preterm birth. Regarding ANC follow-up, 95% of the mothers had attended at least one prenatal visit, and around 96.5% gave birth at health institutions. More than half of the mothers gave birth by spontaneous vaginal delivery while one-third of them delivered by C/S. More than 95% of the delivery was attended by health professionals, and 83.2% of the mothers were delivered single birth. Regarding, chronic medical disease, 58 (14.1%) of the mothers had a history of chronic medical disease, and DM (8.6%), HTN (32.8%), and Anemia (15.5%) were the major chronic disease the mother experienced. Regarding maternal complications, around 150 (36.4%) of the mother experienced one or more than maternal complications. Pregnancy-induced hypertension (PIH) (29.3%), prolonged labor (23.3%), mal-presentation (13.3%) and obstetric hemorrhage (11.3%) were the main maternal complication the mother experienced on their current delivery (Table 2).

### Neonatal characteristics and cause of admission to NICU

The majority of neonates were female (54.8%), and two-third (67.5%) of the neonates were admitted to NICU on the first day of admission. The majority (64.3%) of the neonates were

**Table 1. Socio-demographic characteristics of mothers in WURH and Nekemte Specialized hospital, Nekemte town, western Ethiopia, 2021.**

| Variables | Categories | Frequency | Percent |
|---|---|---|---|
| **Age** | ≤20 | 85 | 20.6 |
| | 21–25 | 150 | 36.4 |
| | 26–30 | 116 | 28.2 |
| | ≥31 | 61 | 14.8 |
| **Residence** | Rural | 163 | 39.6 |
| | Urban | 249 | 60.4 |
| **Marital status** | Never married/single | 57 | 13.8 |
| | Married | 352 | 85.4 |
| | Widowed/divorced | 3 | 0.7 |
| **Educational status** | Unable read and write | 83 | 20.1 |
| | Primary education | 176 | 42.7 |
| | Secondary education | 77 | 18.7 |
| | Tertiary education | 76 | 18.4 |
| **Religion** | Protestant | 242 | 58.7 |
| | Orthodox | 110 | 26.7 |
| | Muslim | 60 | 14.6 |
| **Ethnicity** | Oromo | 365 | 88.6 |
| | Amhara | 44 | 10.7 |
| | Guraghe | 10.6 | 0.7 |
| **Employment** | Housewife | 361 | 87.6 |
| | Gov't employee | 25 | 6.0 |
| | Non gov't employee | 15 | 3.6 |
| | Daily laborer | 11 | 2.7 |
| **Distance of health facility from home** | ≤5 | 148 | 35.9 |
| | 6–10 | 120 | 29.1 |
| | ≥11 | 144 | 34.9 |

hypothermic on admission, and more than half (56.6%) weighted 2500-3900gm at birth. More than one-third of the newborn has initiated breastfeeding within an hour of delivery, and 22.6% had experienced neonatal asphyxia. The main reason for admission was also assessed. Prematurity, birth asphyxia, sepsis/infection, respiratory distress, congenital anomalies such as spinal Bifida and clinical jaundice were the reason for admission to the neonatal intensive unit (Table 3).

## Survival status of neonates

A total of 412 neonates were followed for a median survival time of 27 days with an IQR of 22–28 days. During a follow-up time, a total of 9249 person day observations were detected with a minimum and maximum follow-up time of 1 and 28 days, respectively. Three hundred forty-nine neonates survived at the end of follow-up time and were recorded as censored (Fig 1).

To see the estimate of the survival time, the Kaplan-Meier estimation technique was used. The overall graph of the Kaplan-Meier survivor function indicated a slow decrement of events over a follow-up period. According to this graph, the occurrence of death was high during the first week of the neonatal period. The Kaplan Meier plot for residence shows neonates who were born from a mother of urban residence had a higher survival probability when compared to neonates born from a rural family. The log-rank test showed that there was a significant

**Table 2. Obstetric characteristics and maternal complications of mother delivered in WURH and Nekemte Specialized hospital, Nekemte town, western Ethiopia, 2021.**

| Variables | Categories | Frequency | Percent |
|---|---|---|---|
| **Gestational age** | Preterm | 97 | 23.5 |
| | Term | 294 | 71.4 |
| | Post term | 21 | 5.1 |
| **BMI** | ≤18.4 | 36 | 8.7 |
| | 18.5–25 | 295 | 71.6 |
| | >25 | 81 | 19.6 |
| **Place of delivery** | Home | 15 | 3.6 |
| | Health facility | 397 | 96.4 |
| **Mode of delivery** | SVD | 236 | 57.3 |
| | Instrumental delivery | 51 | 12.4 |
| | C/S | 125 | 30.3 |
| **Number of pregnancy** | 1–2 | 280 | 67.9 |
| | 3–4 | 79 | 19.2 |
| | ≥5 | 53 | 12.9 |
| **ANC follow up** | No ANC follow up | 24 | 5.8 |
| | 1–3 | 225 | 54.6 |
| | ≥4 | 163 | 39.6 |
| **Attendant of delivery** | TBA | 15 | 3.6 |
| | Health professional | 397 | 96.4 |
| **Type of delivery** | Single | 343 | 83.2 |
| | Multiple | 69 | 16.8 |
| **Birth interval** | ≤2 | 303 | 73.5 |
| | 3–4 | 68 | 16.5 |
| | ≥5 | 41 | 10.0 |
| **Chronic medical disease** | No | 354 | 85.9 |
| | Yes | 58 | 14.1 |
| **DM** | No | 53 | 91.4 |
| | Yes | 5 | 8.6 |
| **HTN** | No | 19 | 32.8 |
| | Yes | 39 | 67.2 |
| **Anemia** | No | 49 | 84.5 |
| | Yes | 9 | 15.5 |
| **Pregnancy complications** | No | 301 | 73.1 |
| | Yes | 111 | 26.9 |
| **Maternal complication** | No | 262 | 63.6 |
| | Yes | 150 | 36.4 |
| **Obstetric hemorrhage** | No | 17 | 11.3 |
| | Yes | 133 | 88.7 |
| **Puerperal sepsis** | No | 5 | 3.3 |
| | Yes | 145 | 96.7 |
| **Prolonged labor** | No | 35 | 23.3 |
| | Yes | 115 | 76.7 |
| **PIH** | No | 44 | 29.3 |
| | Yes | 106 | 70.6 |
| **Mal-presentation** | No | 20 | 13.3 |
| | Yes | 130 | 86.7 |

*(Continued)*

**Table 2.** (Continued)

| Variables | Categories | Frequency | Percent |
|---|---|---|---|
| PROM | No | 17 | 11.3 |
| | Yes | 133 | 88.6 |
| Cord prolapse | No | 7 | 4.7 |
| | Yes | 143 | 95.3 |
| Others | No | 10 | 6.7 |
| | Yes | 140 | 93.3 |

ANC- Antenatal care; BMI- Body Mass Index; DM- Diabetes mellitus; HTN-Hypertension, PH- Pregnancy Induced Hypertension

difference in survival probability among the neonates from rural and urban residences (P-value = 0.0286) (Fig 2).

## Incidence density of death and media time to death among neonates admitted to NICU

During the study period, 63 (15.3%) of neonates developed an event or died with an overall incidence rate of death of 6.81(95% CI: 5.32, 8.71) per 1000 person-days observations (PDO). The median time to death was 10 days (95%CI: 5, 19). The incidence rate of death was

**Table 3.** Newborn characteristics of neonates admitted to NICU of WURH and Nekemte Specialized hospital, Nekemte town, western Ethiopia, 2021.

| Variables | Categories | Frequency | Percent |
|---|---|---|---|
| Sex of neonates | Male | 186 | 45.2 |
| | Female | 226 | 54.8 |
| Age of neonates on admission to NICU | ≤ 1 day | 278 | 67.5 |
| | >1 day | 134 | 32.5 |
| Temperature | ≤36.4 | 265 | 64.3 |
| | 36.5–37.5 | 107 | 26.0 |
| | ≥37.6 | 40 | 9.7 |
| Birth weight | <2500gm | 156 | 37.7 |
| | 2500-3900gm | 234 | 56.8 |
| | ≥4000gm | 22 | 5.3 |
| Initiated breast feeding | < 1 hour | 162 | 39.3 |
| | ≥ 1 hour | 250 | 60.7 |
| Respiratory distress | No | 365 | 11.4 |
| | Yes | 47 | 88.6 |
| Asphyxia | No | 319 | 22.5 |
| | Yes | 93 | 77.5 |
| Prematurity | No | 306 | 25.7 |
| | Yes | 106 | 74.3 |
| Infection | No | 268 | 34.9 |
| | Yes | 144 | 65.1 |
| Congenital abnormality | No | 374 | 9.2 |
| | Yes | 38 | 90.8 |
| Jaundice | No | 402 | 2.4 |
| | Yes | 10 | 97.6 |

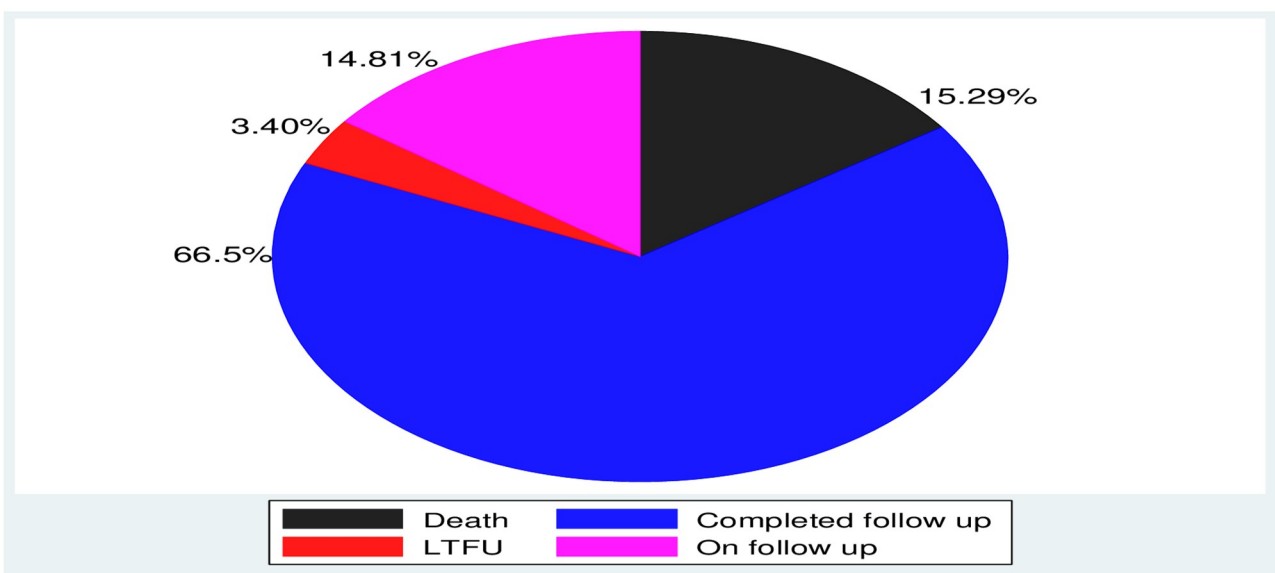

**Fig 1. Survival status of neonates admitted NICU in WURH and Nekemte Specialized hospital, Nekemte town, western Ethiopia, 2021.**

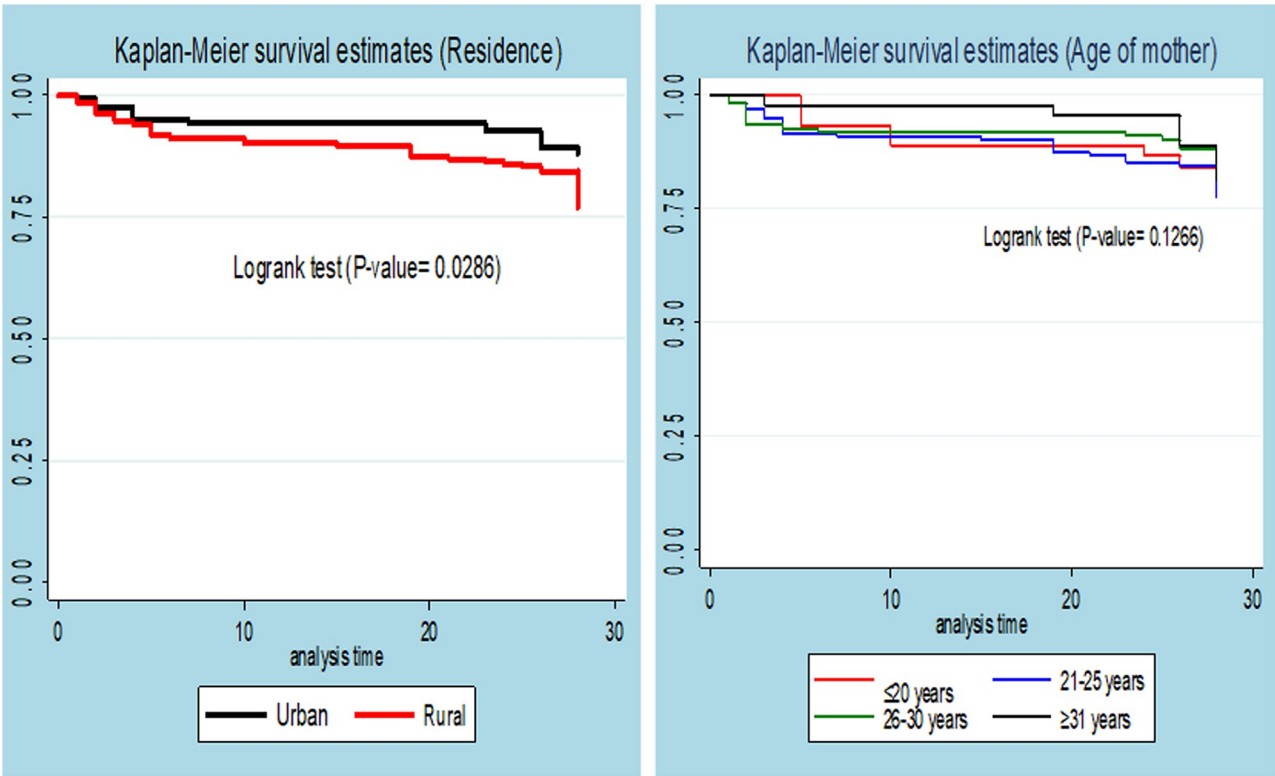

**Fig 2. Kaplan Meier survival curve for residence among neonates admitted to NICU of WURH and Nekemte Specialized hospital, Nekemte town, western Ethiopia, 2021.**

observed at different time intervals and the highest incidence of death was observed during the first week of the neonatal period.

## Predictors of neonatal mortality

In cox regression analysis, residence, age of mother, number of pregnancy, the temperature of neonates, ANC follow up, early initiation of breastfeeding, and gestational age of the mother was selected for multivariable cox regression. Finally, five of the predictors (residence of mother, number of pregnancies, the temperature of neonates, ANC follow up, and early initiation of breastfeeding) were found to have a statistically significant association with neonatal mortality during multivariable cox proportional regression analysis.

Accordingly, the risk of neonatal mortality among neonates delivered from mother of rural settings was 2.04 times higher than neonates born from mother resides in urban settings (AHR = 2.04, 95%CI: 1.14, 3.66). In the same way, the risk of neonatal mortality among neonates delivered from mother having ≤2 and 3–4 number of pregnancy was 59% and 75% lower as compared to neonates delivered from mother of ≥5 number of pregnancy (AHR = 0.41, 95%CI: 0.20, 0.86), (AHR = 0.25, 95%CI: 0.09, 0.69), respectively. The level of neonatal temperature was one of the factors that affect the survival status of neonates. The risk of neonatal mortality was 3.04 times higher among hypothermic neonates than neonates with normal temperature scores (AHR = 3.04, 95%CI: 1.36, 6.80).

ANC visits and early initiation of breastfeeding also predict the survival status of neonates. The risk of neonatal mortality was 7.77 times higher among neonates delivered from mothers who had no ANC visit than their counterparts (AHR = 7.77, 95%CI: 3.99, 15.11). Neonates who have not initiated breastfeeding within the first one hour of delivery were 1.76 more likely to die than those neonates who were initiated breastfeeding within the first hour of delivery (AHR = 2.26, 95% CI: 1.12, 4.56) (Table 4).

## Discussion

Despite a strong commitment of the world community to reduce neonatal mortality, it remains challenging in developing countries like Ethiopia. Hence, this study was aimed to determine survival status, incidence rate and predictors of neonatal death among neonates admitted to the neonatal intensive care unit of Wollega University referral Hospital and Nekemte specialized hospital. The present study has shown the incidence rate of neonatal death was 6.81 per 1000 neonate days' observation with the highest incidence rate of death in the first week of life which was 11.28 per 1000 neonate day's observation. This is a higher incidence rate compared to the study done in Butajira that reported the incidence rate of 1.3 per 1000 neonate days' observation [19]. The possible reason for the observed discrepancy might be due to variations in sample size and the study duration. However, it is a lower incidence rate compared to the study done in southern Ethiopia that revealed the highest incidence rate of neonatal death (27 per 1000 neonate days' observation) [20]. This might be due to the relative improvement of neonatal care before, during, and after delivery in recent times.

Regarding the predictors of neonatal death, the current study found a lack of ANC visits as an independent predictor of neonatal death. The hazard of neonatal death was 7.49 times higher among neonates whose mothers have no ANC visit compared to those who have more than 4 times ANC visit (AHR = 7.49, 95% CI: (3.53, 15.91)). This is also supported with available pieces of evidence from Nigeria, Jimma, Wolaita Sodo University teaching and referral hospital [20–24]. This may be due to the fact that ANC visit helps in the early identification and management of maternal illness and obstetrics complications which may lead to adverse neonatal outcomes. Moreover, this study revealed that the risk of neonatal death increased by

**Table 4. Multivariable analysis of survival status of neonates admitted to NICU, Western Ethiopia, 2021.**

| Variables | Category | Survival status | | CHR | AHR | P-value |
|---|---|---|---|---|---|---|
| | | Dead | Censored | | | |
| **Age of neonates at admission** | ≤ 1 day | 43 | 235 | Ref | ref | |
| | >1 day | 20 | 114 | 0.97 (0.57, 1.65) | 0.79 (0.45, 1.38) | 0.418 |
| **Residence of mother** | Urban | 23 | 226 | Ref | ref | |
| | Rural | 40 | 123 | 2.96 (1.77, 4.95) | 2.04 (1.14, 3.66) | 0.015* |
| **Age of mother** | ≤20 years | 11 | 74 | 1.27 (0.47, 3.45) | 1.60 (0.53, 4.86) | 0.399 |
| | 21–25 years | 30 | 120 | 1.78 (0.74, 4.30) | 2.26 (0.86, 5.92) | 0.097 |
| | 26–30 years | 16 | 100 | 1.19 (0.46, 3.05) | 1.74 (0.64, 4.67) | 0.271 |
| | ≥31 years | 6 | 55 | Ref | ref | |
| **Number of pregnancy** | ≤2 | 42 | 238 | 0.54 (0.30, 0.98) | 0.41 (0.20, 0.86) | 0.019* |
| | 3–4 | 6 | 73 | 0.24 (0.09, 0.629) | 0.25 (0.09, 0.69) | 0.008* |
| | ≥5 | 15 | 38 | Ref | ref | |
| **Temperature of neonate** | <36.5 | 52 | 213 | 3.23 (1.47, 7.12) | 3.04 (1.36, 6.80) | 0.007* |
| | 36.5–37.5 | 7 | 100 | Ref | ref | |
| | >37.5 | 4 | 36 | 1.49 (0.53, 4.18) | 1.57 (0.46, 5.36) | 0.501 |
| **ANC follow up** | No ANC visit | 17 | 7 | 7.93 (4.54, 13.87) | 7.77 (3.99, 15.11) | <0.001* |
| | ANC visit | 46 | 342 | Ref | Ref | |
| **Early initiation of BF** | Within 1 hour | 11 | 151 | Ref | ref | |
| | >1hhour | 52 | 198 | 3.02 (1.57, 5.78) | 2.26 (1.12, 4.56) | 0.022* |
| **Gestational age** | Preterm | 18 | 79 | 1.20 (0.69, 2.08) | 0.88 (0.49, 1.57) | 0.667 |
| | Term | 45 | 279 | Ref | Ref | |
| **Presence of pregnancy complications** | No | 34 | 267 | Ref | ref | |
| | Yes | 29 | 82 | 2.78 (1.69, 4.58) | 1.66 (0.95, 2.89) | 0.071 |

AHR: Adjusted Odds Ratio; CHR: Crude Odds Ratio;

*statistically significant at p<0.05

two folds among neonates who experienced delayed initiation of breastfeeding compared to neonates who received breast milk early (AHR = 2.04, 95% CI: (1.07, 3.89)). This is comparable with the study reports in North Shoa, Amhara regional state, and southern Ethiopia [20, 25, 26]. This is because early breastfeeding initiation helps to prevent hypoglycemia which further helps to reduce the risk of hypothermia and respiratory distress syndrome and eventually improves neonatal survival. Besides, in the present study, nearly one-fourth of the included neonates were preterm babies that might have challenged the early initiation of breastfeeding. Furthermore, early breastfeeding initiation helps to ensure the baby received colostrum which is riched with immunoglobulin to boost the immature immunity of the newborn to reduce the occurrence of severe neonatal infections.

Moreover, the hazard of neonatal death is two-fold higher among neonates whose mothers were from a rural area as compared to their urban counterparts (AHR = 2.20, 95% CI: 1.18, 4.11). This is congruent with a study done in Cameroon [27]. It might be due to the relative variations in access to medical care and health information about home-based neonatal care among rural and urban residents. The risk of neonatal death is 71% & 75% lower among neonates whose mothers had ≤ 2 (AHR = 0.29, 95% CI: 0.13, 0.66) and 3–4 (AHR = 0.25, 95% CI: 0.08, 0.71) pregnancy respectively as compared to mothers who had five or more pregnancy. Existing pieces of evidence also support this finding [28, 29]. This might be due to multi-parity-related obstetric complications leading to bad neonatal outcomes. Probably it might also be due to advanced age-related maternal physiologic degenerations with an increased number of

pregnancies. Though we found no study reporting neonatal body temperature as a predictor of neonatal survival, in the present study, neonatal body temperature predicted neonatal survival. Thus, the risk of neonatal death is threefold higher among hypothermic neonates compared to normothermic neonates (AHR = 2.97, 95% CI: 1.57, 5.60). This is due to the fact that hypothermia is the leading cause of metabolic acidosis and reduced surfactant production causing severe respiratory distress and eventually death. This depicts that most life of newborn babies would have been saved if warm delivery room strategy was maintained, and infections of newborns were identified and treated early.

### Strength and limitations of the study

This study employed a prospective cohort study and advanced statistical model to determine the incidence rate of neonatal death and its predictors. However, there are some limitations with this study. First, this study did not include all important variables such as community and environmental-related factors that might strongly predict neonatal mortality. Second, this study conducted only at the institution level and referral hospitals; we didn't address neonatal mortality at the community level and lower level health institutions (primary hospital and health center) which may underestimate the incidence of neonatal mortality.

## Conclusions and recommendations

The incidence rate of neonatal death was high particularly in the first week of life in the study area. The study found lack of ANC visits, Neonatal hypothermia, increased number of pregnancies, rural residence, and delayed initiation of breastfeeding as independent predictors of neonatal death. Therefore, there is a need to encourage programs that enhance ANC visits for pregnant mothers, and community-based neonatal survival strategies, particularly for countryside mothers. Furthermore, neonates should get special attention during their early neonatal period. Finally, we would recommend future researchers to conduct a study at a community level, and health institutions found at a lower level.

## Supporting information

**S1 File. Dataset.**
(DTA)

**S2 File.**
(DOCX)

## Acknowledgments

We would like to acknowledge Wollega University for covering data collectors and supervisors per diem. Also, our acknowledgement goes to WURH and Nekemte Specialized Hospital for their invaluable co-operation during data collection and our deep acknowledge also for the data collectors, supervisors and health extension worker for their interest and commitment in carrying out the study.

## Author Contributions

**Conceptualization:** Tadesse Tolossa, Bizuneh Wakuma, Getahun Fetensa, Diriba Mulisa, Diriba Ayala, Ilili Feyisa, Ginenus Fekadu, Ayantu Getahun.

**Data curation:** Tadesse Tolossa, Belayneh Mengist, Diriba Ayala.

**Formal analysis:** Tadesse Tolossa, Diriba Mulisa, Haile Bikila.

**Funding acquisition:** Tadesse Tolossa, Bizuneh Wakuma, Belayneh Mengist, Ayantu Getahun.

**Investigation:** Tadesse Tolossa, Getahun Fetensa.

**Methodology:** Tadesse Tolossa, Diriba Mulisa, Diriba Ayala, Haile Bikila.

**Project administration:** Belayneh Mengist, Getahun Fetensa, Ilili Feyisa, Ayantu Getahun.

**Resources:** Bizuneh Wakuma, Diriba Ayala, Ginenus Fekadu, Dube Jara.

**Software:** Tadesse Tolossa, Belayneh Mengist, Diriba Ayala, Haile Bikila.

**Supervision:** Tadesse Tolossa, Getahun Fetensa, Dube Jara.

**Validation:** Ilili Feyisa.

**Visualization:** Ilili Feyisa, Ginenus Fekadu, Ayantu Getahun.

**Writing – original draft:** Tadesse Tolossa, Bizuneh Wakuma, Dube Jara, Haile Bikila.

**Writing – review & editing:** Tadesse Tolossa, Bizuneh Wakuma, Ginenus Fekadu, Dube Jara, Haile Bikila, Ayantu Getahun.

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
