## [Decision Letter · Decision Letter 0]

31 Aug 2021

PONE-D-21-17485

Survival status and predictors of neonatal mortality among neonates admitted to Neonatal Intensive Cara Unit (NICU) of Wollega University referral hospital (WURH) and Nekemte Specialized hospital, Western Ethiopia: a prospective cohort study

PLOS ONE

Dear Dr. Tadesse Tolossa

Thank you for submitting your manuscript to PLOS ONE. After careful consideration, we feel that it has merit but does not fully meet PLOS ONE’s publication criteria as it currently stands. Therefore, we invite you to submit a revised version of the manuscript that addresses the points raised during the review process.

We look forward to receiving your revised manuscript.

Kind regards,

Sarah Saleem

Academic Editor

PLOS ONE

Journal Requirements:

2. Please include additional information regarding the survey or questionnaire used in the study and ensure that you have provided sufficient details that others could replicate the analyses. For instance, if you developed a questionnaire as part of this study and it is not under a copyright more restrictive than CC-BY, please include a copy, in both the original language and English, as Supporting Information. If the original language is written in non-Latin characters, for example Amharic, Chinese, or Korean, please use a file format that ensures these characters are visible.

3. Please state whether you validated the questionnaire prior to testing on study participants. Please provide details regarding the validation group within the methods section.

4. Please amend your current ethics statement to address the following concerns: Please explain why written consent was not obtained, how you recorded/documented participant consent, and if the ethics committees/IRBs approved this consent procedure.

5. Please ensure that you refer to Figure 3 and 7 in your text as, if accepted, production will need this reference to link the reader to the figure.

Additional Editor Comments:

I would like to commend the authors for their efforts in conducting this important study. the manuscript can be improved substantially with hard work. English language needs to be corrected, improved and tightened. Some improvements in the tables are required with clarity in the observed variables for example age to be written as age in completed years, distance in kilometers, the way the chronic medical diseases are given is confusing. categories should be very clear as no chronic condition, Hypertension, Diabetes or more than one comorbidity etc and abbreviations should be avoided. Discussion section is very weak and needs improvement in terms of explaining why babies breastfeeding were not initiated earlier may be mother was sick in ICU which we don't know or baby was premature and sick . The manuscript has the potential to be published but needs major revisions.Hypothermia is given as a cause of death, this needs good discussion as most of the babies who died in earlier ays must be hypothermic. Also data were collected from two hospitals but in discussion only one hospital is mentioned. In the discussion section lots of results are repeated which should be avoided.

Reviewers' comments:

Reviewer's Responses to Questions

**Comments to the Author**

1. Is the manuscript technically sound, and do the data support the conclusions?

Reviewer #1: Yes

Reviewer #2: Yes

2. Has the statistical analysis been performed appropriately and rigorously? 

Reviewer #1: Yes

Reviewer #2: Yes

3. Have the authors made all data underlying the findings in their manuscript fully available?

Reviewer #1: Yes

Reviewer #2: No

4. Is the manuscript presented in an intelligible fashion and written in standard English?

Reviewer #1: Yes

Reviewer #2: No

5. Review Comments to the Author

Reviewer #1: The present study entitled “Survival status and predictors of neonatal mortality among neonates admitted to Neonatal Intensive Cara Unit (NICU) of Wollega University referral hospital (WURH) and Nekemte Specialized hospital, Western Ethiopia: a prospective cohort study” by Tadesse Tolossa at al. is an interesting study with potential public health importance aimed to assess the survival status and predictors of neonatal mortality among neonates admitted to NICU of WURH and Nekemte Specialized Hospital, Western Ethiopia. However, following points should be addressed for strengthening the paper in order to maintain readers’ interest and to convey a clear message.

Major revisions:

• On page 5 lines 26-27, values for P1 and P2 considered for sample size calculation should be reported.

• On page 6 line 1, does the total sample size (396) include lost to follow-up and/or those referred/transferred to other facility?

• On page 6 line 24, Birth weight categories (<2500gm, 2500 - 4000gm, and ≥ 4000gm) are not consistent with categories defined in Table 3 (<2500gm, 2500 – 3900, and ≥ 4000gm).

• Did authors assess confounding and interaction? They should report it on page 8 under data management and analysis.

• Under Ethical Consideration on page 8, authors have mentioned that oral informed consent was obtained whereas about 80% mothers were literate. Was there any specific reason for not obtaining written informed consent? Please specify.

• Result section can be improved further with more appropriate summary of results and by merging table 2 and Figure 1 to make one table of maternal obstetric characteristics and complications, Table 3 and Figure 2 for characteristics of newborn admitted to NICU and the reasons for admission, etc. Similarly, authors should also think about other figures.

• On page 13, line 12 (To see …………..technique was used.) and lines 16 – 19 (A separate graph of…………………..log rank test was computed.) are related to methods and can be moved there.

• Similarly, on page 14, lines 15-16 (Covariates that had P- value ≤ 0.25…………………cox regression analysis) can also be moved methods section.

• Table 4 presents multivariable model results with insignificant predictors (p>0.05). Authors are requested to revisit this multivariable analysis part to present final model results.

• Authors are advised to review list of reference again as some of the references seems incomplete (Refs 5, 9, 14, etc.)

Minor revision

• Some language corrections will improve the overall quality of paper.

Reviewer #2: This is a critical area of study where data is either scarce or unavailable. This study methodology has several major flaws:

1. This study was carried out in two hospitals. I believe a hospital description, such as the number of beds in each ward and incubators or ventilators in the NICU, annual births, NICU admissions, mortality rate, and so on, would be beneficial. You also mentioned two other hospitals (Marie Stopes International and FGAE Hospital) - did these facilities take part in the study? If not, there is no need to mention it.

2. Incomplete sample size assumptions (you should mention the percent of neonatal mortality among exposed and the percent of mortality among unexposed) Because there are many exposures, you must calculate sample size for each exposure and the highest sample size should be used for this study. Furthermore, you did not specify how much the sample size was inflated due to lost to follow-up/withdrawal from the study.

3. You must define variables as needed. Birth asphyxia, prematurity, sepsis, and so on.

4. "Data collection procedure" should be used instead of "Data collection tools and techniques."

5. Please explain why oral consent was given.

6. I suggest redo the analysis using standard categories. for example, term and preterm, ANC (no ANC vs. at least one ANC visit)

7. Figures 1 and 2 could be useful if categorized by baby outcome status (alive/dead).

8. Figure 4 is unnecessary.

9. What is the average length of survival?

10. Please check for grammatical and punctuation errors and strongly advise having the manuscript reviewed by an English native speaker.

6. PLOS authors have the option to publish the peer review history of their article (what does this mean?). If published, this will include your full peer review and any attached files.

Reviewer #1: No

Reviewer #2: No

---

## [Author Response · Author response to Decision Letter 0]

12 Sep 2021

Dear   Assistant Editor-in-Chief PLOS ONE

Dear Editor, this is regarding the entitled as “Survival status and predictors of neonatal mortality among neonates admitted to Neonatal Intensive Cara Unit (NICU) of Wollega University referral hospital (WURH) and Nekemte Specialized hospital, Western Ethiopia: a prospective cohort study” submitted to PLOS ONE. Thanks for your time and consideration in editing the manuscript. We have carefully read your comments and corrected inline of your comments and suggestions. All comments raised were edited and incorporated in the main manuscript. Some of the changes were highlighted with yellow color in the manuscript

Response: Thank you Dear editor, we accepted your comment. All the revision was made in line with the journal requirements including the figure.

2. Please include additional information regarding the survey or questionnaire used in the study and ensure that you have provided sufficient details that others could replicate the analyses. For instance, if you developed a questionnaire as part of this study and it is not under a copyright more restrictive than CC-BY, please include a copy, in both the original language and English, as Supporting Information. If the original language is written in non-Latin characters, for example Amharic, Chinese, or Korean, please use a file format that ensures these characters are visible.

Response: Dear editor, we have included questionnaire used in the study as a supplementary file 

3. Please state whether you validated the questionnaire prior to testing on study participants. Please provide details regarding the validation group within the methods section.

Response: Thank you dear editor, the tool was not validated on study participants. However, to maintain the reliability and validity of the tool we have checked its consistency by translating from English language to local language, than retranslated back to ENGLISH by expert of the language. 

4. Please amend your current ethics statement to address the following concerns: Please explain why written consent was not obtained, how you recorded/documented participant consent, and if the ethics committees/IRBs approved this consent procedure.

Response: Thank you dear editor. I know this very important question; however, written consent was not obtained from patient/care givers of the neonates. In Ethiopia, written consent is only possible if a sample is needed from the patients and any invasive procedure is performed. Since we were not received any blood sample and invasive procedure were not performed, only verbal consent was obtained from mothers.

5. Please ensure that you refer to Figure 3 and 7 in your text as, if accepted, production will need this reference to link the reader to the figure.

Response: revised

Additional Editor Comments:

I would like to commend the authors for their efforts in conducting this important study. the manuscript can be improved substantially with hard work. English language needs to be corrected, improved and tightened. Some improvements in the tables are required with clarity in the observed variables for example age to be written as age in completed years, distance in kilometers, the way the chronic medical diseases are given is confusing. categories should be very clear as no chronic condition, Hypertension, Diabetes or more than one comorbidity etc and abbreviations should be avoided. Discussion section is very weak and needs improvement in terms of explaining why babies breastfeeding were not initiated earlier may be mother was sick in ICU which we don't know or baby was premature and sick . The manuscript has the potential to be published but needs major revisions.Hypothermia is given as a cause of death, this needs good discussion as most of the babies who died in earlier ays must be hypothermic. Also data were collected from two hospitals but in discussion only one hospital is mentioned. In the discussion section lots of results are repeated which should be avoided.

Response: Thank you dear editor, we have answered your question in the discussion part of revised manuscript

Reviewers' comments:

Reviewer's Responses to Questions

Comments to the Author

Review Comments to the Author

Reviewer #1: The present study entitled “Survival status and predictors of neonatal mortality among neonates admitted to Neonatal Intensive Cara Unit (NICU) of Wollega University referral hospital (WURH) and Nekemte Specialized hospital, Western Ethiopia: a prospective cohort study” by Tadesse Tolossa at al. is an interesting study with potential public health importance aimed to assess the survival status and predictors of neonatal mortality among neonates admitted to NICU of WURH and Nekemte Specialized Hospital, Western Ethiopia. However, following points should be addressed for strengthening the paper in order to maintain readers’ interest and to convey a clear message.

Response: Dear reviewer, thank you for taking a time to review our work thoroughly. We have tried to incorporate your comments in the main manuscript. In addition, here we have answered your questions. Hope we have addressed all your concerns

Major revisions:

• On page 5 lines 26-27, values for P1 and P2 considered for sample size calculation should be reported.

Response: Thank you Dear reviewer, we have accepted your comment and incorporated in the revised manuscript

• On page 6 line 1, does the total sample size (396) include lost to follow-up and/or those referred/transferred to other facility?

Response: Thank you Dear, actually 398 sample size was calculated by adding 10% non-response rate. It did not include lost to follow up and neonates referred for further treatment. They were included under censored data.

• On page 6 line 24, Birth weight categories (<2500gm, 2500 - 4000gm, and ≥ 4000gm) are not consistent with categories defined in Table 3 (<2500gm, 2500 – 3900, and ≥ 4000gm).

Response: Thank you dear, we have corrected it

• Did authors assess confounding and interaction? They should report it on page 8 under data management and analysis.

Response: Dear reviewer, thank you very much for this important question. Sorry, we have not considered interaction effect in our analysis, but we have controlled the effect of confounding by fitting multivariable analysis.

• Under Ethical Consideration on page 8, authors have mentioned that oral informed consent was obtained whereas about 80% mothers were literate. Was there any specific reason for not obtaining written informed consent? Please specify.

Response: Thank you dear editor. I know this very important question; however, written consent was not obtained from patient/care givers of the neonates. In Ethiopia, written consent is only possible if a sample is needed from the patients and any invasive procedure is performed. Since we were not received any blood sample and invasive procedure were not performed, only verbal consent was obtained from mothers.

• Result section can be improved further with more appropriate summary of results and by merging table 2 and Figure 1 to make one table of maternal obstetric characteristics and complications, Table 3 and Figure 2 for characteristics of newborn admitted to NICU and the reasons for admission, etc. Similarly, authors should also think about other figures.

Response: Thank you dear, we have merged figures ij table 2 and 3 as per your recommendations

• On page 13, line 12 (To see …………..technique was used.) and lines 16 – 19 (A separate graph of…………………..log rank test was computed.) are related to methods and can be moved there.

Response: Revised

• Similarly, on page 14, lines 15-16 (Covariates that had P- value ≤ 0.25…………………cox regression analysis) can also be moved methods section.

Response: Revised

• Table 4 presents multivariable model results with insignificant predictors (p>0.05). Authors are requested to revisit this multivariable analysis part to present final model results.

Response: Dear reviewer, we are very much grateful for your critical look into our manuscript. Regarding the model, we have raised a series of steps followed during analysis for building and selecting best fitting model in the previous reviewer’s comment. Firstly, we selected variables for multivariable analysis based on their bivariable P-value (0.25), secondly, we started to build model through backward selection method and thirdly, we selected the best model from the result of backward stepwise selection using log likelihood ratio (LLR). Here, we selected model one among the eight models built during the backward selection process because this model has relatively higher value of statistics. Obviously it is known that the higher the value of statistics, the better the model in Log likelihood ratio. 

• Authors are advised to review list of reference again as some of the references seems incomplete (Refs 5, 9, 14, etc.)

Response: Revised

Minor revision

• Some language corrections will improve the overall quality of paper.

Response: Dear reviewer, we are thankful for your important comment and we have tried to edit the grammatical flaws throughout the manuscript in its revised version. We have edited the spelling, grammatical errors, incomplete and poorly structured sentences throughout the manuscript. 

Reviewer #2: This is a critical area of study where data is either scarce or unavailable. This study methodology has several major flaws:

1. This study was carried out in two hospitals. I believe a hospital description, such as the number of beds in each ward and incubators or ventilators in the NICU, annual births, NICU admissions, mortality rate, and so on, would be beneficial. You also mentioned two other hospitals (Marie Stopes International and FGAE Hospital) - did these facilities take part in the study? If not, there is no need to mention it.

Response: Thank you dear reviewer, we have accepted your comment and included in the revised manuscript

2. Incomplete sample size assumptions (you should mention the percent of neonatal mortality among exposed and the percent of mortality among unexposed) Because there are many exposures, you must calculate sample size for each exposure and the highest sample size should be used for this study. Furthermore, you did not specify how much the sample size was inflated due to lost to follow-up/withdrawal from the study.

Response: Thank you dear reviewer, we have incorporated sample size issue in the revised manuscript

3. You must define variables as needed. Birth asphyxia, prematurity, sepsis, and so on.

Response: Thank you dear, we have accepted your comment and incorporated in the revised version of manuscript

4. "Data collection procedure" should be used instead of "Data collection tools and techniques."

Response: thank you dear, we have modified it

5. Please explain why oral consent was given.

Response: Thank you dear editor. I know this very important question; however, written consent was not obtained from patient/care givers of the neonates. In Ethiopia, written consent is only possible if a sample is needed from the patients and any invasive procedure is performed. Since we were not received any blood sample and invasive procedure were not performed, only verbal consent was obtained from mothers.

.

6. I suggest redo the analysis using standard categories. for example, term and preterm, ANC (no ANC vs. at least one ANC visit)

Response: Thank you dear, we have revised according to your comment

7. Figures 1 and 2 could be useful if categorized by baby outcome status (alive/dead).

Response: Dear reviewer, we believe your comment is worthwhile in improving the manuscript but we have removed figure 1 and 2 as per recommendation of reviewer 1.

8. Figure 4 is unnecessary.

Response: removed

9. What is the average length of survival?

Response: Thank you dear reviewer, the median survival time for overall neonates was 27 days and median time to death for event (death) was 10 days.

10. Please check for grammatical and punctuation errors and strongly advise having the manuscript reviewed by an English native speaker.

Response: Dear reviewer, we are thankful for your important comment and we have tried to edit the grammatical flaws throughout the manuscript in its revised version. We have edited the spelling, grammatical errors, incomplete and poorly structured sentences throughout the manuscript. Now we believe the revised version is clean and clear enough to the readers.

---

## [Decision Letter · Decision Letter 1]

14 Apr 2022

PONE-D-21-17485R1Survival status and predictors of neonatal mortality among neonates admitted to Neonatal Intensive Care Unit (NICU) of Wollega University referral hospital (WURH) and Nekemte Specialized hospital, Western Ethiopia: a prospective cohort studyPLOS ONE

Dear Dr. Tolossa,

Thank you for submitting your manuscript to PLOS ONE. After careful consideration, we feel that it has merit but does not fully meet PLOS ONE’s publication criteria as it currently stands. Therefore, we invite you to submit a revised version of the manuscript that addresses the points raised during the review process.

 The manuscript has been seen by two reviewers, and their comments are appended below. In addition your manuscript has been seen by staff editors at PLOS ONE, and there are several outstanding issues by which the article does not meet our publication criteria. Please see these comments here:

PLOS journals require authors to make all data underlying the findings described in their manuscript fully available without restriction, with rare exception. PLOS journals will not consider manuscripts for which authors will not share data because of personal interests, such as patents, commercial interests or potential future publications (https://journals.plos.org/plosone/s/data-availability).

- The values behind the means, standard deviations and other measures reported.

- The values used to build graphs

Authors must share the “minimal data set” for their submission. PLOS defines the minimal data set to consist of the data required to replicate all study findings reported in the article, as well as related metadata and methods. Additionally, PLOS requires that authors comply with field-specific standards for preparation, recording, and deposition of data when applicable.

As it stands, the manuscript does not meet our data availability requirements. Please either upload the underlying data as supplementary files to the manuscript, or deposit this in an appropriate public repository and indicate where this data will be available upon publication.

In the abstract you mention: 'Studies are limited in Ethiopia in general, and no study has been conducted in the study area in particular'. However, we were able to find several studies on the topic:

https://pubmed.ncbi.nlm.nih.gov/34512074/

https://pubmed.ncbi.nlm.nih.gov/34040481/

https://pubmed.ncbi.nlm.nih.gov/34512075/

https://pubmed.ncbi.nlm.nih.gov/31987037/

https://pubmed.ncbi.nlm.nih.gov/32238144/

https://pubmed.ncbi.nlm.nih.gov/27499702/

Please update the rationale for performing the study in the Abstract, given the citations above. Please outline why it is necessary to perform the study is the specific area studied in the Introduction to provide a rationale for your study.

Can you please revise the manuscript to address the concerns raised above?

We look forward to receiving your revised manuscript.

Kind regards,

Sebastian Shepherd

Associate Editor

PLOS ONE

Reviewers' comments:

Reviewer's Responses to Questions

**Comments to the Author**

1. If the authors have adequately addressed your comments raised in a previous round of review and you feel that this manuscript is now acceptable for publication, you may indicate that here to bypass the “Comments to the Author” section, enter your conflict of interest statement in the “Confidential to Editor” section, and submit your "Accept" recommendation.

Reviewer #1: All comments have been addressed

Reviewer #2: All comments have been addressed

2. Is the manuscript technically sound, and do the data support the conclusions?

Reviewer #1: Yes

Reviewer #2: Yes

3. Has the statistical analysis been performed appropriately and rigorously? 

Reviewer #1: Yes

Reviewer #2: Yes

4. Have the authors made all data underlying the findings in their manuscript fully available?

Reviewer #1: Yes

Reviewer #2: No

5. Is the manuscript presented in an intelligible fashion and written in standard English?

Reviewer #1: Yes

Reviewer #2: Yes

6. Review Comments to the Author

Reviewer #1: (No Response)

Reviewer #2: (No Response)

7. PLOS authors have the option to publish the peer review history of their article (what does this mean?). If published, this will include your full peer review and any attached files.

Reviewer #1: No

Reviewer #2: No

---

## [Author Response · Author response to Decision Letter 1]

21 Apr 2022

Dear   Assistant Editor-in-Chief PLOS ONE

Dear Editor, this is regarding the entitled as “Survival status and predictors of neonatal mortality among neonates admitted to Neonatal Intensive Cara Unit (NICU) of Wollega University referral hospital (WURH) and Nekemte Specialized hospital, Western Ethiopia: a prospective cohort study” submitted to PLOS ONE. Thanks for your time and consideration in editing the manuscript. We have carefully read your comments and corrected inline of your comments and suggestions. All comments raised were edited and incorporated in the main manuscript. Some of the changes were highlighted with yellow color in the manuscript

Editor comment

In the abstract you mention: 'Studies are limited in Ethiopia in general, and no study has been conducted in the study area in particular'. However, we were able to find several studies on the topic:

Response: Thank you dear, we have accepted your comment and addressed in the revised manuscript. Yes a number of researches were conducted on this area, but the main gap is that the previous studies were limited to extracting data from patient cards, and only their focus was the survival status of neonates during the seven days of life or while they are in admission room. Only limited studies were conducted by following neonates for 28 days.

---

## [Editor Report · Decision Letter 2]

9 May 2022

Survival status and predictors of neonatal mortality among neonates admitted to Neonatal Intensive Care Unit (NICU) of Wollega University referral hospital (WURH) and Nekemte Specialized hospital, Western Ethiopia: a prospective cohort study

PONE-D-21-17485R2

Dear Dr. Tolossa,

We’re pleased to inform you that your manuscript has been judged scientifically suitable for publication and will be formally accepted for publication once it meets all outstanding technical requirements.

Kind regards,

George Vousden

Staff Editor

PLOS ONE

Additional Editor Comments (optional):

The updated manuscript indicates that "In addition previous studies were only conducted when the neonates were in the admission unit by overlooking the survival status after the neonates discharged from hospital" please add 'were' as follows (the added text should not be bolded in the manuscript): "In addition previous studies were only conducted when the neonates were in the admission unit by overlooking the survival status after the neonates **were** discharged from hospital"
---

## [Editor Report · Acceptance letter]

11 Jul 2022

PONE-D-21-17485R2 

Survival status and predictors of neonatal mortality among neonates admitted to Neonatal Intensive care Unit (NICU) of Wollega University referral hospital (WURH) and Nekemte Specialized hospital, Western Ethiopia: a prospective cohort study 

Dear Dr. Tolossa:

I'm pleased to inform you that your manuscript has been deemed suitable for publication in PLOS ONE. Congratulations! Your manuscript is now with our production department. 

Kind regards, 

on behalf of

Dr. George Vousden 

Staff Editor

PLOS ONE